# Relations between Circular Economic "Principles" and Organic Food Purchasing Behavior in Hungary

**Csaba Fogarassy** [1], **Kinga Nagy-Pércsi** [2,*], **Sinazo Ajibade** [3], **Csaba Gyuricza** [4,*] **and Prespa Ymeri** [5]

[1] Climate Change Economics Research Centre, Szent István University, Páter Károly st. 1, 2100 Gödöllő, Hungary; Fogarassy.Csaba@gtk.szie.hu

[2] Institute of Regional Economics and Rural Development, Szent István University, Páter Károly st. 1, 2100 Gödöllő, Hungary

[3] Rural Development Engineering Course Program, Szent István University, Páter Károly st. 1, 2100 Gödöllő, Hungary; jacobssinazo@gmail.com

[4] National Agricultural Research and Innovation Centre, Institute of Crop Production, Szent Istvan University, Szent-Györgyi Albert utca 4, 2100 Gödöllő, Hungary

[5] Doctoral School of Management and Business Administration, Szent István University, 2010 Gödöllő, Hungary; Ymeri.Prespa@phd.uni-szie.hu

* Correspondence: Nagyne.Percsi.Kinga@gtk.szie.hu (K.N.-P.); Gyuricza.Csaba@mkk.szie.hu (C.G.)

**Abstract:** Because of the climate change and emerging need for an environmentally sustainable production system, circular economic characteristics have come to the front in many studies. There are many challenges in this shift toward a circular value chain. Still, it is unquestionable that the analysis of consumers' behaviour is crucial, because without their engagement, circular systems cannot work correctly. This article aimed to explore the circular characteristics of consumers' attitude towards food purchasing in Hungary. Factor and cluster analyses were applied for market segmentation. The question to be answered was the following: "Are there any segments in the Hungarian food market that can be aimed at by different marketing tools to promote circular systems?" The hypothesis was that well-defined segments can be separated, garnering more engagement in the circular value chain in Hungary. We could separate two clusters, in which the members' opinions were in line with the circular economic characteristics. Summing up the features of the different clusters, we can state that the members in cluster 1 ("Information-dependent") and cluster 3 ("Direct purchasers") were in the most local dimension; their attitude was the most adequate for the circular economic values. The "Information-dependent" consumer in particular was remarkable from the aspect of this investigation. This study showed that highly educated young people, who are very conscious consumers and live on good incomes, may be the target group for circular innovation. These young consumers usually buy organic food, are confident internet and software users, live in cities, and follow a healthy lifestyle. Finding the right marketing tools to integrate these consumers into more sustainable circular systems effectively and to be committed to the concepts of circular consumption is an essential mission in the future. Collecting from different databases and continuously analysing consumer feedback can be a huge step towards in achieving sustainable consumption and avoiding food waste. The significance of this analysis was that we found a defined segment that represents propensity towards accepting circular economy values and can be the target group of policies integrating circular systems.

**Keywords:** circular value chain; sustainability; circular innovation; consumers' attitude; direct food purchasing; short food supply chains; local food system

## 1. Introduction

The circular economy concept, which is gaining momentum and attention from industries, policymakers, and academia [1,2], has a high interest in reduction, reuse, and recycling of resources [3]. The drivers for the transition to a circular economy is the challenge faced by the global agricultural food production, as well as the estimated population increase of 9.6 billion (UN, 2017) by 2050 putting global agriculture under a large amount of pressure due to decreasing arable land, rising urbanization, and extreme climatic conditions due to global warming. Thus, producers are pressurized to increase crop yields using environmentally friendly agricultural practices, while perceiving natural resources and feeding a growing population [4]. However, in order to meet the expectations of the circular economy, innovative and modern technology that allows for the recovery of valuable materials should be established [5,6]. This is one of the reasons the circular economy is frequently promoted as an environmentally friendly way to facilitate green economic growth and entrepreneurial opportunities. However, for environmental management, one of the global challenges is to ensure that all the activities conform to sustainable development principles [7], creating a balance of three social, economic, and ecological aims, as stated by Krajnc and Glavič [8]. The pillars of sustainability are included in many topics, from social to agricultural sciences, and other components have been added to the three well-known supports of sustainability. For example, Khwidzhili and Worth [9] stated that in order to have a sustainable agricultural production system, it must first meet the requirements of biological productivity, meet the protection of natural resources, be economically viable, reduce the level of risk, and be socially acceptable. Interestingly, the revised five pillars are by the objectives of the circular economy, and according to Lilja [10], the adequate concept to seek global sustainability is a local circular economy approach.

It is important to note that the organic-based natural treatment generates nutrient-rich substrate in the circular economy, which are used in the production of food as they are non-toxic, environmentally friendly, and economical. For instance, it is stated that the primary goal of the circular economy is to recycle organic materials completely into the primary resources [11–13]. Organic agriculture is a resilient model of circular economic principles. Thus, the sector has received closer focus, as a large number of nations, companies, and organisations are promoting its use. Organic agriculture is characterised as a holistic approach, as it considers the long-term environmental sustainability by producing food in an environmentally friendly manner that sustains soil health, the ecosystem, and the people [14–17]. Nevertheless, consumer requirements for organic products can be a challenge in the circular economy, as the need for the products and services can remain low when the behaviour changes. Another reason is that the adaption of efficient use of natural resources and the changing processes of business requires the involvement of consumers [18]. The consumers in wealthy nations find organic agriculture to be a better option for climate protection, animal welfare, and to be environmentally friendly [19,20]. This can be due to the positive public image of organic agriculture, which is touted as a concept for sustainable agriculture [21]. Concerns about the effect of detrimental inorganic fertiliser on human health and the environment can also be a reason for consumer growing demand for organic products [22]. Nevertheless, this is an issue in developed nations because most consumers are aware of the health benefits of organic farming [23]. However, consumption of organic food has potential in the circular economy (CE) because short food supply chains are connecting to organic food production and are used by the consumers efficiently, representing locally locked production.

In order to meet the circular economy, several concerns need to be taken into consideration by the actors involved in this transformation [24]. These issues matter to the society and, in particular, the position of consumers. Furthermore, there are new, innovative technologies applied in the circular economy, which is in contrast to the naturalness of organic food production. These are novel foods, and due to different reasons, the consumers are reluctant to change their eating and purchasing habits and are not committed to taking part in these systems [25]. There are different reasons that consumers try to avoid foods that are related to novel technologies, most of which are economical and psycho-behavioural factors, as they have no control over new technology, and a sense of unnaturalness can also occur;

thus, it is challenging for them to see the benefits, status quo bias, and other factors [26]. Different studies show that the use of the socio-technical perspective in consumption helps to understand the role of consumers in the circular economy better. The research highlighted the importance of a rapid transition to a circular economy for the domestic sector.

The significance of focusing on the activities that represent local life is reinforced by the view that different strategies towards reaching "circularity" are already in progress in households. In the circular economy, consumer collaboration is an essential part; this calls for a new and more active involvement of consumers [17,27], while at this time it is still not enough due to the lack of environmental awareness and interest in the CE. It is an interesting question as to whether organic food consumers can be engaged in the circular economy and as to who would be the future consumers of food products stemming from circular systems, as well as asking what rules can be used to support the development of the consumer suppling systems.

## 2. Literature Review and Theoretical Background

Consumers play an important role in the transition towards circular economy through making more sustainable choices and promoting them further. Sustainable policies and practices should be promoted, which may lead to the emergence of some initiatives [28]. Chamberlin and Box [29] identified 10 groups of factors from the circular economy and sustainability literature that may affect consumers' acceptance of circular economy products and services; however, none of them were related specifically with organic food consumption and circular economy. Although the food system has been highlighted as a potential site for the successful implementation of the "cycle" principles, much of the work focuses solely on food production and the reuse of surplus food, with less emphasis on food-related consumption, except end-of-production or waste [28,30–32]. It should be mentioned however, that there are an increasing number of studies recently that have underlined the importance of consumers and consumption in the further development of a CE [33]. Consumers can have different roles in a CE. They can act as purchasers, maintainers, repairers, sellers, sharers, collaborators, and waste discarders [34]. Some authors point out that the active involvement of consumers in achieving CE goals is in contrast to the current situation characterized by lack of consumer awareness and acceptance [35–38]. There are authors who have suggested conducting more research on the type of individuals or groups that are more susceptible to accepting circular solutions [39,40], and related to this, exploring strategies to improve the acceptance of policy [39,41]. These also underpin the relevance and importance of the identification and analyzation of the possible target groups among the consumers. To find the possible target groups who can be committed to CE principles, the markets should be segmented on the basis of different aspects relating to circularity. However, insights into consumer perception about the CE as the full concept are not available [27]. For this reason, other indices can be used in analysis. Among others, zero-kilometre, organic, small amount, and trust are aspects that are important in a CE [17]. The activity of the target groups can be triggered by a combination of certain motifs and characteristics of more sustainable products and services [27]. It is also challenging for the policy to instigate changes in consumption patterns because many people feel that dietary choices are a private matter, and that their freedom of choice would be impinged upon by governmental dietary recommendations [28]. Relating to this, it is also an interesting question as to what the role of the media would be in creating conditions for a transition to a circular economy [33]. Many consumers are not aware of the entire food chain, the various actors involved, and the moral implications associated with their decisions. A consumer environment in which people make routine choices these days does not promote more sustainable consumption patterns [28].

The results show that young adults are primarily concerned with purchasing and the circumstances of the environmental friendly purchase, and less concerned for the process of sustainable use and disposal in their consumer behaviour. The main barriers identified are high price, lack of information, as well as missing knowledge and abilities [42]. Other studies point out that consumption processes are different with regard to gender [43], age, and income [44]. Moreover, some studies

show that the sustainable consumption behaviour of young adults is less distinct than those of older people. With regard to food consumption, when young adults (14–17 years old) buy food on their own, they prefer cheap and tasty takeaways without truly considering the production line. Some prefer organic food, but primarily for their own health, not for ecological or social reasons [44]. Additionally, there are findings that show that young adults are virtually connected and communicate via social media. In doing so, they are essentially able to search for missing information on products, social work conditions, or disposal programs, and are able to share this information [45]. With regard to sustainable consumer behaviour, the results show that there were no significant differences in age, gender, and budget [42,46]. In addition, food consumption habits are related to people's value orientation, emotions, personal and collective identity, traditions, and eating culture. Meat consumption, for example, is linked to certain frameworks of masculinity and also to ethnicity [47]. According to the abovementioned information, it can be assumed that organic consumers have propensity towards accepting circular economy values because the main motivations for eating organic food are the positive health impact, environmental protection, and animal welfare considerations in many countries [48–53]. CE solutions require consumers to integrate new products, re-use existing infrastructures, or register themselves in a completely different way, for example by adopting consumer service systems [54–56]. Thus, those consumers who are open-minded and ready to use innovative solutions can be involved.

In organic farming there are different innovative purchasing systems and direct sales forms that are used by the increasing number of the consumers. The solidarity purchasing groups and community-supported agriculture also operate in different countries successfully [17,57]. It is clear to everyone that there is much potential for sustainability in short supply chains, provided they meet the right economic, environmental, and social conditions. However, Born and Purcell [58] emphasize that "local traps" should be avoided, which means local systems should not be automatically declared as "good practices", because "local food" is not equal to "sustainable food" [59]. The circular economic model is based on the prudent and prudent use of resources to reduce the environmental burden in this way. This requires a right attitude on the part of producers and a shift in consumers' food purchasing habits towards sustainability. It can be a tool, for example, to support low-carbon footprints or food mile distance products, as well as to act consciously to avoid wasting food and to reduce waste [17]. The potential positive effects of short supply chains on the circular economy and sustainability goals can be realized by favouring local food and small farmers. This may take the form of a significant reduction in waste and, inter alia, the emergence of greater trust between producers and consumers. According to the literature review, some important aspects have been selected and used in analysis to characterize the different consumer groups. These aspects are reflected from the chosen statements of the consumer attitude and other relevant features among the consumers' behaviour. The preference of local food and small-scale farmers represents propensity towards accepting circular economy values. The impact of local food to the environment is questionable if we think of the consumers travelling to the site of the farmers. This problem was mentioned also by Kiss et al. [17]. Maybe in this case, the requirement of 0 miles is lost, but it fulfils the circular requirements because the consumer would like to know the product and the production circumstances consciously. As for organic food, sometimes it can come up against difficulties. In the case of the global organic food market, consumption is separated and located far from production. However, consumption of organic food has potential in the circular economy (CE) because short food supply chains are connected to organic food production, which are used by the consumers efficiently and represent locally locked production. With the help of innovative solutions of CE applied in organic farming and other environmentally friendly production systems, the old, bad innervation (fear of fake organics) can be avoided.

## 3. Material and Methods

The research was carried out on the largest Hungarian organic markets (Biokultúra Organic Markets) in February 2018. After applying five pilot questionnaires, the questionnaire was improved on

the basis of useful experiences gained with the other 31 questionnaires that were made through personal interviews. The interviewees had the chance to give their judgment regarding specific questions. We used the aspects discussed in the literature review, especially the aspects used by Kiss et al. [17] to analyse the receptivity of the circular values among the Hungarian consumers. The survey focused on the aspects of trust between consumers and producers, the health-consciousness of the consumers, and the use of the direct channels that are contributed to by the loops (producer-consumer interactions) and forming of loops in CE. In the Hungarian food economy, there are no examples to investigate, and thus we can analyse only the receptivity of CE values from the side of the consumers.

### 3.1. Sampling and Survey Instrument

The questionnaire had 16 questions, which can be grouped in terms of consumer behaviour, attitudes, eating habits, and factors influencing consumers' purchase decisions, purchasing channels, and judgment of food safety over the demographic features of the interviewees. Generally, the survey came into existence on the basis of Hungarian professional literature, which was related to the objectives of the article. The consumers' attitude was assessed using 10 statements, such as "I can buy safer food in the market." (Safer market food), "I try to buy food made by small scale farmers." (Preference of small-scale farmers), "I found the activity of the authorities adequate.", "I always read the name of the processor on the label." (Reading label), "I prefer Hungarian food." (Hungarian food), "I buy only trusty food." (Searching trusty food), "I try to purchase healthy food.", "I think I am a conscious consumer." (Conscious consumer), "I trust in food sold by food stores." (Food from food store), "Most of the food is full of harmful ingredients." (Harmful ingredients). The scale was a five-point Likert scale, which was anchored at "1", indicating strong disagreement and "5", indicating strong agreement. Not all of these statements were used in the analysis because we had to exclude some of them and use some other food purchase decision factors that had circular aspects and were supported by our survey. These were the label on the food product, the ingredients, the nutrition value, the good health impact, the origin, and the additive content of the food. We used the Likert scale in the case of other questions of the survey also. After we conducted spot questionnaires as it is described above, we collected 811 additional questionnaires with the help of students who were participating in the courses of "Food safety and quality assurance" and "Hygiene in catering" in Szent István University. Students had to make a questionnaire to one of their family members or friends until the end of March 2018. The requirement was that interviewees need to be above 18 years old. Finally, 842 questionnaires were collected this way. The response rate was 98.34% from 842 questionnaires, but only 828 were suitable to investigate the behaviour and characteristics of food consumers. Unfortunately, we had to exclude 14 participants from further analysis because of too much missing data or a high level of inconsistency. There were 18 respondents who did not buy organic food. However, they used the organic market as a food purchasing channel, and thus we decided to process their data also. This problem could be due to misunderstanding the concept, as in Hungary, the official name for this type of food is "ecological" (which means organic), but numerous consumers recognise them as "bio food".

### 3.2. Data Analysis

Descriptive statistical analysis was used for the study, and in order to segment the organic consumers, we used factor and cluster analysis. We used the SPSS software, version 24, to analyse the data. Factor analysis was performed, and segmentation was conducted using *k*-means cluster analysis. The factor scales consisting of four factors were used in the cluster analysis. Before *k*-means clustering, a hierarchical cluster analysis using Ward linkage was performed to identify the adequate number of clusters. After conducting the analysis, four clusters were obtained (Figure 1). By using the mean of consumer attitude scores for each cluster, the differences between segments were considered. In Table 1, we can see the variables that were used for segmentation; in order to determine the smallest number of meaningful factors, factor analysis using principal axis factoring and Varimax rotation was

used. Bartlett's test of sphericity was significant at the 0.001 level, and the Kaiser–Meyer–Olkin (KMO) value was higher than 0.7 [60].

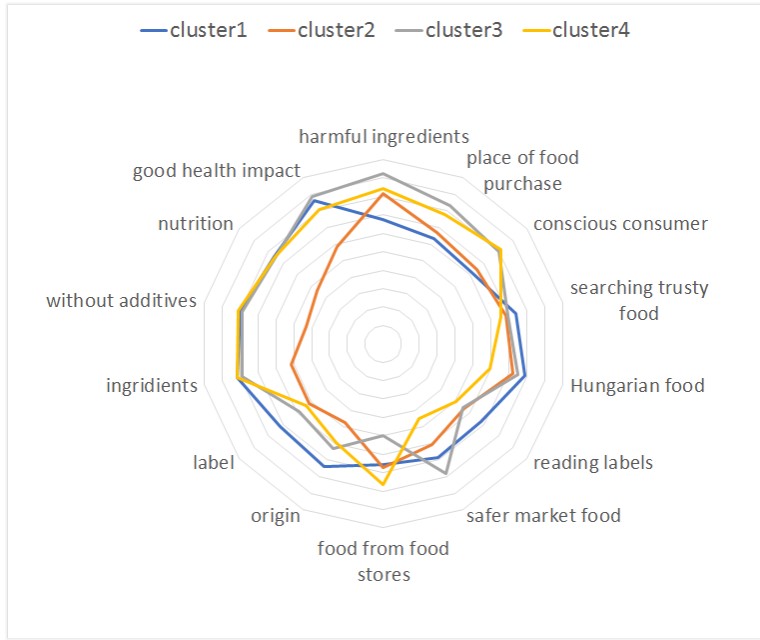

**Figure 1.** Images of each cluster (cluster 1, cluster 2, cluster 3, cluster 4).

**Table 1.** Variables used for segmentation.

| Variables | Mean | Standard Deviation |
|---|---|---|
| Harmful ingredients | 4.11 | 0.971 |
| Place of purchase | 3.67 | 1.099 |
| Conscious consumer | 3.64 | 1.112 |
| Searching trusty food | 3.48 | 1.077 |
| Hungarian food | 3.58 | 1.135 |
| Reading labels | 2.87 | 1.248 |
| Safer market food | 3.21 | 1.170 |
| Food from food stores | 3.19 | 0.939 |
| Origin | 3.06 | 1.351 |
| Label | 2.96 | 1.274 |
| Ingredients | 3.68 | 1.199 |
| Without additives | 3.55 | 1.228 |
| Nutrition | 3.44 | 1.190 |
| Good health impact | 3.96 | 1.114 |

The abovementioned cluster 1, together with cluster 3 (*N* = 249), were the most innovative and open-minded clusters. The members in cluster 2 (*N* = 195; "Careless") cared the least about the ingredients, nutritional value, and health impact of the food (Figure 1). They trusted in food sold by food stores and did not think that they can buy safer food in the market. They were not interested in reading labels and in the origin of the food.

The members in cluster 3 intensively used the direct food purchasing channels also (Figure 2), similarly to members in cluster 1. The biggest difference compared to the other groups was that the organic market had the highest preference in these two groups and that they used other direct forms with higher frequency (Figure 2).

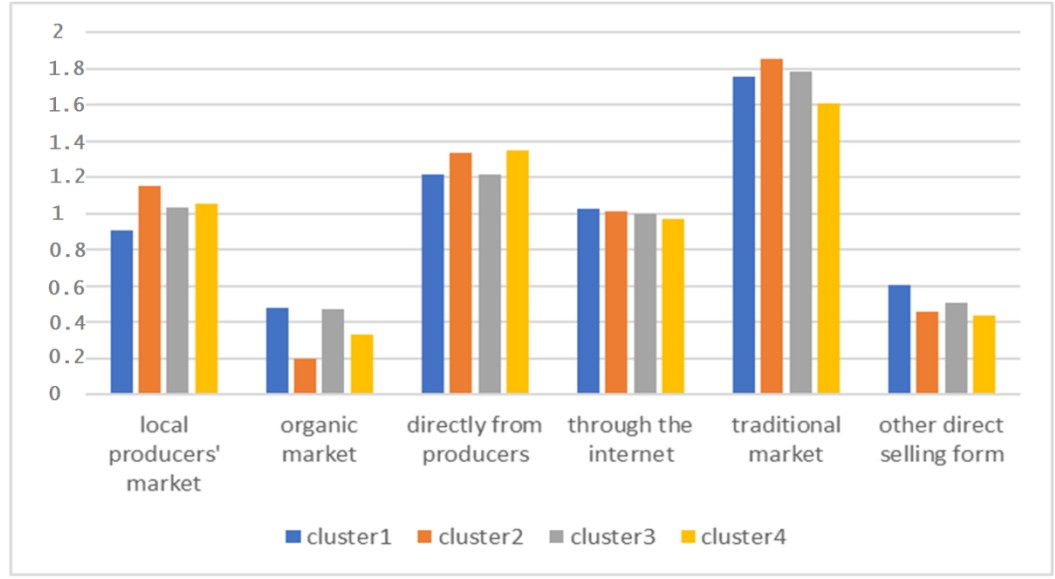

**Figure 2.** Channels frequently used in different segments.

## 4. Results and Discussion

The factor analysis resulted in 14 observed variables allocated to four factors. According to the data in Table 2, component 1 contains those variables that relate to the components of food. Component 2 refers to the origin of food and food tracking. Component 3 refers to the consciousness of the consumers. It consists of the selection of purchasing channel, the fear of harmful ingredients, and the self-assessment of the consumer. Component 4 represents the market purchase as the counterpart of store purchase. The four factors were named "Food components"; "Tracking", where the label on the product plays an important role in food purchasing; "Consciousness"; and finally, "Market purchase vs. store purchase".

**Table 2.** Rotated component matrix.

| Items | Component | | | |
|---|---|---|---|---|
| | **1** | **2** | **3** | **4** |
| Most food is full of harmful ingredients. | 0.097 | −0.036 | **0.725** | 0.201 |
| I take care of that where I buy food. | 0.189 | 0.308 | **0.807** | 0.012 |
| I think I am a conscious consumer. | 0.263 | 0.265 | **0.818** | −0.068 |
| I buy only trustworthy food. | 0.116 | **0.565** | 0.183 | 0.03 |
| I prefer Hungarian food. | 0.069 | **0.722** | 0.135 | 0.224 |
| I always read the name of the processor on the label. | 0.132 | **0.722** | 0.169 | −0.024 |
| I can buy safer food in the market. | 0.063 | 0.271 | 0.05 | **0.733** |
| I trust in food sold by food stores. | −0.039 | 0.076 | −0.075 | **−0.821** |
| Origin | 0.451 | **0.566** | 0.043 | 0.051 |
| Label | 0.318 | **0.58** | −0.028 | −0.003 |
| Ingredients | **0.759** | 0.241 | 0.2 | −0.029 |
| Without additives | **0.818** | 0.142 | 0.156 | 0.022 |
| Nutrition | **0.713** | 0.093 | 0.09 | 0.049 |
| Good health impact | **0.687** | 0.229 | 0.159 | 0.107 |
| Cronbach's alpha | 0.798 | 0.720 | 0.780 | −0.775 |

Extraction method: principal component analysis. Rotation method: Varimax with Kaiser normalization. Rotation converged in eight iterations. (Clusters were formed based on bold values.).

Finally, four clusters could be separated on the basis of the factor and the hierarchical cluster analyses. We identified those characteristics that can have a strong relation to circular economic principles, such as "I can buy safer food in the market." Regarding these features, we selected the segments, which can be the purpose group for the circular ideas and systems. The most crucial characteristic of members of the cluster 1 ($N = 194$; "Information dependent") that they used. The most important purchase influencing factors for the individuals were the positive health impact, the ingredients, and that the food contained no additives. The origin of the food in the purchasing decision process and the preference of Hungarian food products were the most important in this cluster, as compared to the other groups. Their food purchase was also influenced by the label, and consequently, they searched for trustworthy food. They also read the name of the producer on the food label ("Information dependent"). They seemed to be very conscious consumers who are inclined to buy food in the market. They were found to be highly qualified, and they belonged to a higher income category. They also took care of the wrapping of food and had an inclination to buy food from small farmers, with this latter point being caused by the searching for quality food, and thus they identify small farmers' products with higher quality or in terms of social responsibility.

Cluster 3 ($N = 249$) named "Direct purchasers" consisted of middle-aged people (Figure 3) who were mainly employees from different parts of the country (Figure 4). They had a fear of harmful ingredients, but they were not influenced by the label very much. However, as compared to the other groups, label and origin were important influencing factors in their purchasing decisions. They preferred Hungarian food also. The biggest difference to the other groups was that they thought they can buy safer food in the market, and thus they preferred direct contact with the sellers or producers. This factor had the highest impact on their attitudes.

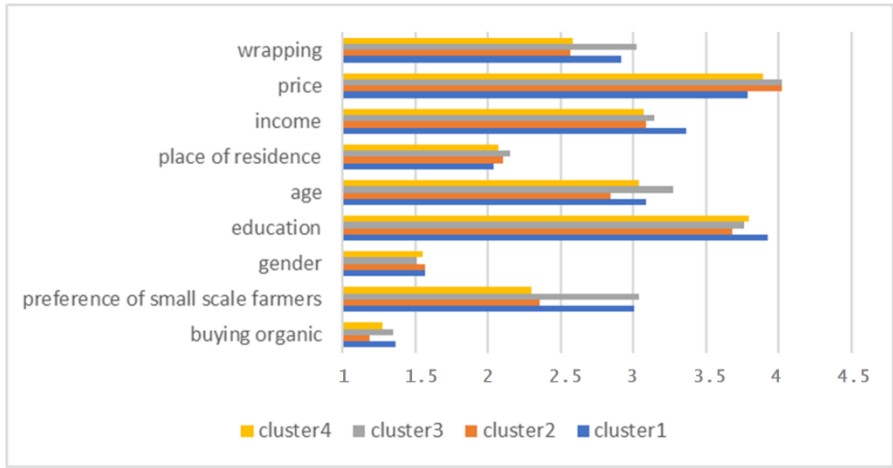

**Figure 3.** Demographic characteristics of the clusters. Gender: 1—male; 2—female; income 1–5, 3 was the average; Education 1–5, 1—primary, 2—vocational, 3—technical, 4—grammar, 5—college or university. In the case of buying organic, there were to options to choose: 1 meant "I don't buy organic food." and 2 meant "I buy organic food.".

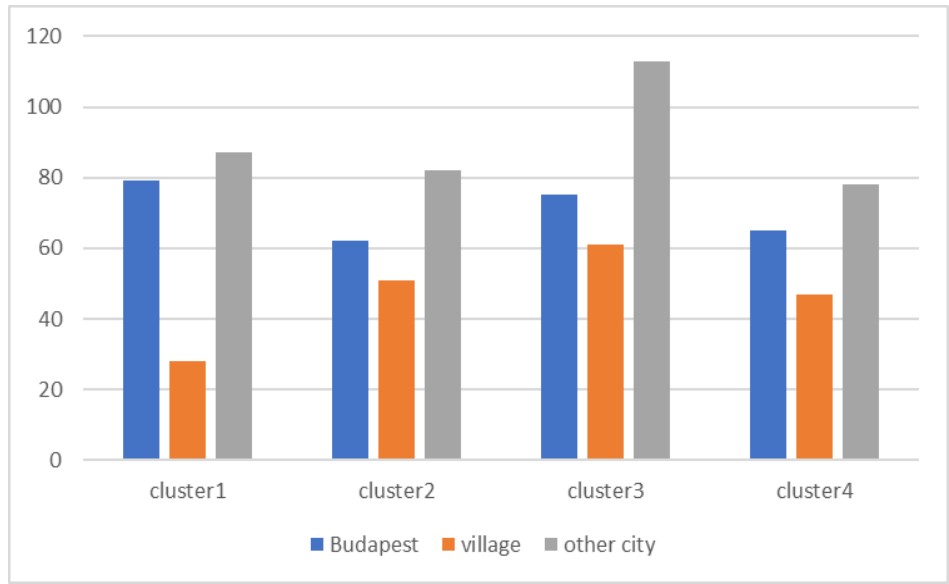

**Figure 4.** Distribution of respondents by place of residence.

The members of cluster 4 (*N* = 190; "Food store fans") were the most conscious consumers according to their opinion. They would rather trust food sold in food stores than food sold in markets. They payed much attention to the ingredients and other components of food, and they were not influenced by the trademark and food producer in food purchase. Most of them were highly qualified (Figure 3) and lived in cities (Figure 4).

Members in clusters 1 and 3 were the most qualified and preferred to buy organic food also (Figure 3). Most of the members of these two clusters were employees (Figure 5).

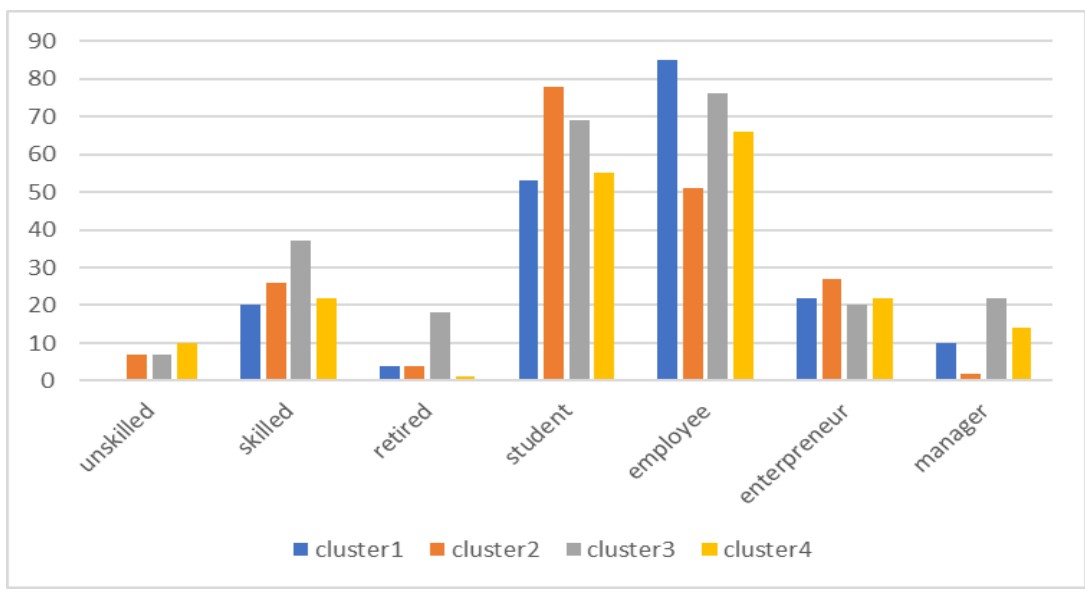

**Figure 5.** Distribution of respondents by occupation.

For the 21–30 age group, well-considered, environmentally friendly purchasing was a highlight, but it is was more important for them to avoid wasting or to engage in conscious waste management. For other age groups, these areas were equal (Figure 6).

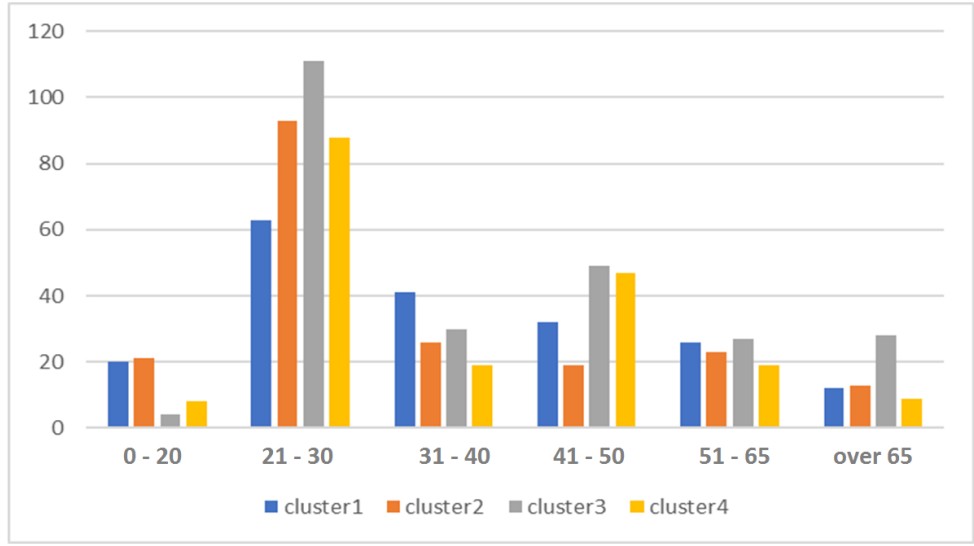

**Figure 6.** Distribution of respondents by age. (Age groups are shown in bold.).

The elements from the attitude determining factors that could have a strong relation to the circular concept were selected. Although the idea of short supply chains is not based on waste reduction, they can contribute to the prevention of food waste, and in this way, to the objectives of a circular economy. Due to this reason, the direct connections and the frequent use of short supply chains refer to circular values. The inclination to buy small farmers' and local or domestic products also follows circular economic concepts [17]. To investigate which cluster has more potential in being engaged in CE, three indices were formulated with the help the selected elements in accordance with those mentioned before. The statements that were the nearest to the circular values were the following: (a) "I prefer Hungarian food.", (b) "I always read the name of the processor on the label.", (c) "I can buy safer food in the market.", and (d) "I try to buy food made by small-scale farmers." Only one was suitable for selection in the case of linear characteristics, namely, (e) "I trust in food sold by food stores.".

Index 1 contains the average scores given by the consumers for the selected circular characteristics: (c + d)/(c + d + e). Index 2 is a ratio for the circular characteristics and is calculated as follows: (a + b + c + d)/(a + b + c + d + e). These ratios were calculated for the whole sample also. As we can see in Figure 7, the highest circular score was achieved in clusters 1 and 3, and these two clusters had the strongest preferences to direct channels.

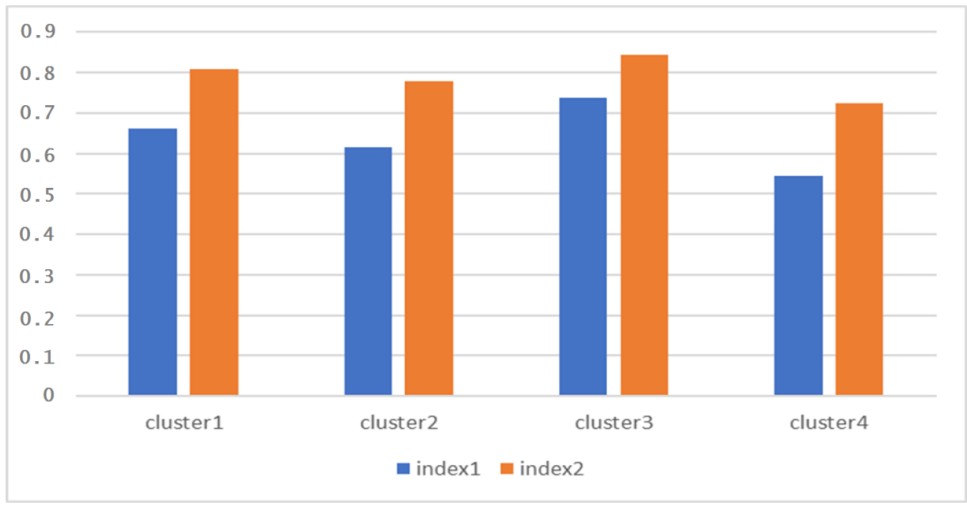

**Figure 7.** Rate of the circular values in the different clusters: Index 1 is a ratio for direct food purchasing.

According to the research results and the literature, the consumers with high salaries and the consumers in rich nations believe that organic farming is a better choice for climate protection, animal welfare, and the environment. This may be due to the positive image of organic farming, which is recognized in the concept of sustainable agriculture. Concerns about the impact of harmful inorganic fertilizers on human health and the environment may also be the reason for increasing demand for organic products. In the most advanced countries, consumers are aware of the health benefits of organic farming, and consumer behaviour regarding the willingness to buy organic products depends on the country (or regions) under study. Organic food is primarily consumed in the local production and consumption system, which is one of the basic principles of the circular economy, where short food supply chains are connected, and producers and consumers know each other's needs. However, there are new, innovative technologies used in the circular economy, contrary to the natural nature of organic food production. For various reasons, consumers are reluctant to change their eating and shopping habits and are not committed to participating in the systems (online shopping, platforms, box system, home delivery, etc.) [55]. Waste avoidance in a circular economy is an essential part of functioning systems for consumer cooperation. This requires the active involvement of consumers through new and modern means. The circular value chain is not created by itself, it requires the combined presence of many effects. It is clear that modern ICT (Information and Communications Technology) tools are the key to transformation in the linear circular shift, being primarily digital-enabled systems.

## 5. Conclusions

It is essential to discover and analyse consumer behaviour and attitudes, as without their commitment, the circular economy, referred to as a synonym for sustainability, cannot function. According to this survey, we can separate two clusters (cluster 1 of young people and cluster 2 of older people) in which the members' opinions are in line with the circular economic characteristics. Summing up the features of the different clusters, we can state that the members in clusters 1 and 3 were in the most local dimension, and their attitude was the most adequate in terms of the circular economic values. They had the most definite preference for direct relations with the consumers. Cluster 1 in particular was remarkable from the aspect of this investigation. They were highly qualified young people who were very conscious consumers with good income circumstances. They usually bought organic food and lived in a city. It is a very important mission for future research to find the proper marketing tools, the help of which can be integrated into the systems, becoming committed to the CE concept. From our point of view, clarifying the link between food consumption and food waste is a crucial area for the introduction of circular systems. The quantity of wasted food, associated with more expensive and better quality food, is significantly lower than with linear (global) or traditional food production–consumption. Food waste can be significantly reduced by creating loops in the bio-economy; however, we do not currently have accurate data. Of course, the features and speed of the process relies on consumer decisions. The fundamental building blocks of the circular economy, such as sharing databases, up-to-date information for everyone, introducing short supply chains, correct and detailed knowledge of market players, and customising consumption, enhance the value creation process in consumer communities. The introduced research means the starting point, the base of a next survey where we can use the experiences gained from this one. It would also be interesting to find the answer to the question as to whether the organic food consumers are more engaged in circular technologies and methods, or whether they refuse the new innovative, digital, and circular value chain food supply systems. Foreseeable trends indicate that the major consumers of organic food will change significantly (younger and higher-income consumers will take the lead), and the volume of consumption may increase significantly due to digitization. Our research also found that the 40–65 age group, which today plays a leading role in the consumption of organic food, is much more inflexible in using the key elements of the circular value chain, and that their preference for local foods is not as strong as that for the conscious younger (and more digital) generation. The results of

the research mainly show that a group of consumers (mainly in the age group of 20–30 years) with high income and relatively little information about organic products has appeared on the market of organic products. On the basis of the cluster classification, this group was named "Information-dependent", which indicates, among other things, that it makes its decisions on the basis of the information available to it, primarily digital information. It is important for them to read everything if they can access it, but only if it is available in digital form. The building blocks of the circular economy (sharing and service platforms, smart operation, industrial symbiosis, big data using, etc.) originate from and feed back into digital economic systems, and thus predictable, ecological consumption systems will have to proceed this way in the future.

**Limitations:** Our study did not cover all possible topics, but was limited to what we considered important. The research did not represent the consumer habits of organic food in Hungary, but the results identified the dominant tendencies of the consumer community. Generalization of research was only possible within a limited framework.

**Author Contributions:** Conceptualization, review, editing, and supervision—C.F.; conceptualization, investigation, methodology, and formal analysis—K.N.-P., C.G.; resources and writing—S.A.; methodology and formal analysis—P.Y. All authors have read and agreed to the published version of the manuscript.

**Funding:** This research received no external funding.

**Acknowledgments:** Preparation the manuscript and our final article was supported by the Climate Change Research Centre and Doctoral School of Management and Business Administration at Szent István University.

**Conflicts of Interest:** The authors declare no conflict of interest.

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
