# Peer review of "Relations between Circular Economic “Principles” and Organic Food Purchasing Behavior in Hungary"

_agronomy, doi:10.3390/agronomy10050616_

Round 1

Reviewer 1 Report

The authors have addressed the reviewer's comments. Please polish the writing to prevent redundancies, grammatical errors and punctuation problems.

Author Response

Dear Reviwer1, 

Thank you very much for your valuable comments earlier, which were very helpful in finalizing the article.

The Authors

Reviewer 2 Report

The authors have addressed all my comments and I have no further comments.

Author Response

Dear Reviwer2, 

Thank you very much for your valuable comments earlier, which were very helpful in finalizing the article.

The Authors

Reviewer 3 Report

The following responses relate to my original comments and your subsequent responses.

Comment one:

The introduction is significantly improved. The research question stated in the abstract should be the same as the question in the introduction. Currently, the question in the introduction is different to the question stated in the abstract.

Comment two:

The literature review is much improved. You have provided an empirically based description of the literature which gives the reader a good idea of the state of the extant literature. Inclusion of some relevant customer engagement theoretical frameworks would also have been desirable.

Comment three:

The methodology is also improved, however, it is important to establish the validity of survey instrument for the purpose, whereas you have really just described it.

Comment four:

The current analysis presentation provides a descriptive analysis of the data. I acknowledge that the indices you have identified may be useful, however, had you inserted some customer engagement theoretical into the analysis, this could have provided a relational, rather than a categorisational result which may have been very interesting.

Comment five:

I agree that buying from the farmer is indicative of a circular economy, but it is not equivalent.

Author Response

Dear Reviewer3,

Thank you very well for valuable comments and notes. It is a great honor that you have accurately read our article thoroughly again. We will provide the following answers to your comments. 

Comment one: The introduction is significantly improved. The research question stated in the abstract should be the same as the question in the introduction. Currently, the question in the introduction is different to the question stated in the abstract.

Answer:Thanks for the comment, I supplemented the previous question, clarified it according to the comment! I added the next addition: "..what rules can be used to support the further development of the consumer system?" - I indicated the addition in red in the text!

Comment two: The literature review is much improved. You have provided an empirically based description of the literature which gives the reader a good idea of the state of the extant literature. Inclusion of some relevant customer engagement theoretical frameworks would also have been desirable.

Answer:Thank you for your comment and comment. We did not want to highlight this direction in the present discussion. In the next consumer survey, which we have already launched, we will focus on these issues.

Comment three: The methodology is also improved, however, it is important to establish the validity of survey instrument for the purpose, whereas you have really just described it.

Answer:We are aware of the limitations and validity of the research. I added the next addition to Limitations part: "Generalization of research is only possible within a limited framework."

Comment four: The current analysis presentation provides a descriptive analysis of the data. I acknowledge that the indices you have identified may be useful, however, had you inserted some customer engagement theoretical into the analysis, this could have provided a relational, rather than a categorisational result which may have been very interesting.

Answer:Thank you very much for your kind comment. As before, we will discuss this issue in great detail in the next survey, the results of which will be reported in detail in our next scientific paper.

Comment five: I agree that buying from the farmer is indicative of a circular economy, but it is not equivalent.

Answer: I agree with your comment. The direction we present indicates the possibility and the direction related to the use of ITC tools. But we didn’t want to underscore this very much, because the circular economy alone is basically about the development of that direction.

Thank you also very much for your valuable comments earlier, which were extremely helpful in finalising the article!

Best regards,

Csaba Fogarassy

This manuscript is a resubmission of an earlier submission. The following is a list of the peer review reports and author responses from that submission.

Round 1

Reviewer 1 Report

This paper studies the relations between circular economic "principles" and organic food purchasing behavior. I agreed to review this article as I am interested in circular economy, which targets on waste mitigation and environmental sustainability. Circular economy is more sustainable as products will be consumed and “disposed of” close to the site of their manufacturing, where “one person’s waste = another’s raw material” (TerraInfirma, 2013). Unfortunately, we didn't read much analysis and findings on circular economy in this article. The authors are suggested to improve their analysis to reveal the relations between circular economy and organic food; or how the organic food industry can stimulate circular economy.

Author Response

Dear Reviewer 1,
Thank you very much for your comment, because your question points out very well the direction from which it is necessary to explain the criteria previously used for sustainable consumption. The main purpose of the article is to map the consumer changes in the Hungarian organic food market, to examine what changes can be forecast in the future. We looked at how the basic elements of a circular economic concept defined as a new discipline of sustainability can affect this product range.
Based on your suggestions, we rewrote the abstract and expanded the literature review section.
We were able to significantly improve the quality of the article after rethinking its related suggestions. Thanks for the additional thoughts!
The Authors

Reviewer 2 Report

Comments:

  1. This manuscript aimed to explore the circular characteristics of consumers' attitude towards food purchasing in Hungary. Although, this study addresses an interesting topic with practical utility, it needs further improvement.
  2. Lines 15-20: these lines read too general with many warm-up sentences introducing the topic. The main objective of the study should be mentioned earlier, and the authors should keep 2-3 warm-up sentences.
  3. The main objectives should be emphasized in the abstract. Currently, it is too general and fails to show the main contribution of the study to the topic.
  4. Line 24: “Our studies show that highly educated young…”; it is preferred not use personal pronouns (e.g. I, we, our) in the journal-articles. This sentence should be revised to: this study showed that…”
  5. The authors should highlight the main conclusion of the study at the end of the abstract in 1-2 sentences as well.
  6. Overall, the authors should outline the significance of their study in the abstract in 1-2 sentences.
  7. Lines 39-40: the authors should add relevant references to support their arguments in these lines.
  8. The Introduction section is very lengthy, yet it failed to reflect and discuss the main problem in a critical way. In other words, this section belongs to the critical debates on the topic and highlighting other similar studies’ findings.
  9. Most inputs in the Introduction section (lines 55-157) should be moved to a new section “Theoretical background”.
  10. Figure 1 is not well-designed; the authors should improve this figure considering aesthetics aspects.
  11. To enrich the Introduction section, the authors should add a few relevant/recent studies along with their approaches and outcomes and indicate the main contribution of the current study by comparing it with previous ones.
  12. My main concern with this paper is on the extent to which the study makes an original and significant contribution of the field. Currently, I find the paper's contribution unclear.
  13. To complete my previous comment, the authors need to identify the “gap” in the literature, make a compelling argument that why the “gap” needs to be filled, and provide the readers with explanations on appropriate and practical approaches.
  14. More importantly, the main topic is not discussed very properly in the context of Hungary; The reader should be introduced to more about the context in relation to the impacts of organic farming and the challenges posed to Hungarian agricultural food production. Some statistics and data would be helpful.
  15. What about the main objectives? The authors should clearly discuss the main goal and specific objectives of their study at the end of the Introduction section.
  16. In the Methods section, to avoid a descriptive format, the authors should summarize the questions (lines 195-201) in a table.
  17. In the Result and discussion section, the authors should add more explanations on the contributions of Figures 5-9.
  1. The Result and discussion section should include the explanation and extension of the results and further explain the limitation of the study (lines 360-362 should be moved to this section).
  1. Moreover, this section needs further enrichment as the discussion of the results according and compared to existent literature is missing. The authors should outline how the main findings are in line with previous studies.
  2. The authors should add more elaboration on the main implications (especially policy implications) of findings in the Conclusion section.
  3. What about future research directions? This item should be also outlined in the Conclusion section.
  4. The whole text should be improved in terms of English and style (especially the narrative format and personal pronouns).

Author Response

Dear Reviewer2,
Thank you very much for your helpful comments. Based on your suggestions, we have reorganized the content to a considerable extent and the individual chapters have been rewritten. We changed the logical order and put the content units together much more strongly. Detailed answers to the comments can be found here:

you can find the reviwer's comments with bold letters and the authors' answers with normal letters:

1.This manuscript aimed to explore the circular characteristics of consumers' attitude towards food purchasing in Hungary. Although, this study addresses an interesting topic with practical utility, it needs further improvement.
2. Lines 15-20: these lines read too general with many warm-up sentences introducing the topic. The main objective of the study should be mentioned earlier, and the authors should keep 2-3 warm-up sentences.
3. The main objectives should be emphasized in the abstract. Currently, it is too general and fails to show the main contribution of the study to the topic.

Based on the reviewers' suggestions, we revised the first part of the paper completely, adding 26 more references to the list, and left out more references during the conversion. A new Literature review chapter has been added, which took into account these details. Thank you very much for your comments on re-structuring and setting goals.

4. Line 24: “Our studies show that highly educated young…”; it is preferred not use personal pronouns (e.g. I, we, our) in the journal-articles. This sentence should be revised to: this study showed that…”

Has been changed, corrected! 

5. The authors should highlight the main conclusion of the study at the end of the abstract in 1-2 sentences as well.

6. Overall, the authors should outline the significance of their study in the abstract in 1-2 sentences.

Thanks for your comment, the Abstract of the paper has been improved!

"This article aimed to explore the circular characteristics of consumers' attitude towards food purchasing in Hungary. Factor and cluster analyses were applied for market segmentation. The question to be answered is the following: “Are there any segments in the Hungarian food market, which can be aimed at by different marketing tools to promote circular systems to?”. The hypothesis is that well-defined segments can be separated, garnering more engagement in the circular value chain in Hungary. We could separate two clusters in which the members' opinions are in line with the circular economic characteristics. Summing up the features of the different clusters, we can state that the members in Cluster 1 (“Information dependent”) and Cluster 3 (“Direct purchasers”) are in the most local dimension; their attitude is the most adequate to the circular economic values. Especially the “Information dependent” consumer is remarkable from the aspect of this investigation. This studies show that highly educated young people, who are very conscious consumers, and live on good incomes, may be the target group of circular innovation. These young consumers usually buy organic food, are confident Internet and software users, live in cities, and follow a healthy lifestyle. Finding the right marketing tools to integrate these consumers into more sustainable circular systems effectively and to be committed to the concepts of circular consumption is an essential mission in the future. Collecting from different data bases and continuously analysing consumer feedback can be a huge step towards achieving sustainable consumption, and avoiding food waste. The significance of this analysis is that we could find a defined segment which represents propensity towards accepting circular economy values and can be the target group of the policy to integrate in circular systems."

7. Lines 39-40: the authors should add relevant references to support their arguments in these lines.

8. The Introduction section is very lengthy, yet it failed to reflect and discuss the main problem in a critical way. In other words, this section belongs to the critical debates on the topic and highlighting other similar studies’ findings.

9. Most inputs in the Introduction section (lines 55-157) should be moved to a new section “Theoretical background”.

Based on your suggestion we have added a new references on independent literature review chapter. New texts are marked with red letters throughout the document. The first part of the thesis was completely rewritten. Some of the circular principles are presented in the literature review, but the purpose of the paper is not to present the building blocks of circular systems. A further 26 new publications were added to the references and several references were omitted

10. Figure 1 is not well-designed; the authors should improve this figure considering aesthetics aspects.

Has been deleted from the paper. 

11. To enrich the Introduction section, the authors should add a few relevant/recent studies along with their approaches and outcomes and indicate the main contribution of the current study by comparing it with previous ones.

A new literature review has been added to the paper. Further references.

12. My main concern with this paper is on the extent to which the study makes an original and significant contribution of the field. Currently, I find the paper's contribution unclear.

We have expanded the section on conclusions and explained the details. Thank you very much for your comment.

13. To complete my previous comment, the authors need to identify the “gap” in the literature, make a compelling argument that why the “gap” needs to be filled, and provide the readers with explanations on appropriate and practical approaches.

Very few literature has so far investigated the link between organic food consumption and circular systems. Our contribution to the research area is to analyze current consumer habits on the basis of circular aspects (trust, organic consumption, health awareness, preference for local produce, conscious purchase) already formulated by other authors (Kiss et al., 2019). We have segmented the domestic consumer market by appropriate consumer attitudes and behaviors and have found a segment that can be addressed by policy makers.

14. More importantly, the main topic is not discussed very properly in the context of Hungary; The reader should be introduced to more about the context in relation to the impacts of organic farming and the challenges posed to Hungarian agricultural food production. Some statistics and data would be helpful.

It was not important for the purpose of the article to introduce the Hungarian system of organic farming, because it operates in the EU system and the conditions are the same as in other EU countries.
Based on the proposal, we have added a source of references to provide further information on the state of organic farming in Hungary.

Nagy-Pércsi, K.; Fogarassy, C. Important Influencing and Decision Factors in Organic Food Purchasing in Hungary. Sustainability 2019, 11, 6075.

Kiss, K.; Ruszkai, C.; Takács-György, K. Examination of Short Supply Chains Based on Circular Economy and Sustainability Aspects. Resources 2019, 8, 161.

15. What about the main objectives? The authors should clearly discuss the main goal and specific objectives of their study at the end of the Introduction section.

Has been corrected and improved: 

"The study highlighted the importance of a rapid transition to a circular economy for the domestic sector. The significance of focusing on the activities that represent local life is reinforced by the view that different strategies towards reaching ‘circularity’ are already in progress in households. In the circular economy, consumer collaboration is an essential part; this calls for a new and more active involvement of consumers while at this time is still not enough due to lack of environmental awareness and interest in CE. It is an interesting question that organic food consumers can be engaged in the circular economy or who would be the future consumers of food products stemming from circular systems?"

16. In the Methods section, to avoid a descriptive format, the authors should summarize the questions (lines 195-201) in a table.

To data analysis chapter we have added a table which is a clear description of the variables. 

17. In the Result and discussion section, the authors should add more explanations on the contributions of Figures 5-9.

Thank you for your comment, we have added few more sentences to explain the different Figures. 

18. The Result and discussion section should include the explanation and extension of the results and further explain the limitation of the study (lines 360-362 should be moved to this section).
19. Moreover, this section needs further enrichment as the discussion of the results according and compared to existent literature is missing. The authors should outline how the main findings are in line with previous studies.

Has been expanded wit more explanations and additional sentences. 

20. The authors should add more elaboration on the main implications (especially policy implications) of findings in the Conclusion section.

Has been added policy implications also! 

21. What about future research directions? This item should be also outlined in the Conclusion section.

The conclusion section has been improved with that part as well. Future researches are needed on the field of the consumers’ familiarity with CE principles and values. How deep do these CE elements (waste avoidance, environment protection, sustainability, healthy diet) root in the Hungarian society. The new generation is a so called “Youtube – generation” and it is a positive evolvement that there are Hungarian Youtube channels focusing on “no waste lifestyles” created by young generations. 

22. The whole text should be improved in terms of English and style (especially the narrative format and personal pronouns). 

We improved the narrative format in the text. Thank you for your notes! 

In closing, thank you very much for your valuable comments. Based on your proposals, the content of the thesis has been importantly expanded. The quality of the article changed significantly after the corrrections.
The Authors

Reviewer 3 Report

Introduction:

Your introduction moves too much around between marketing circular economy to a particular group of customers, operations improvement, economic impact and sustainability goals. All are relevant, however, the sequence of argument is jumbled and discursive, making the theoretical position of the paper unclear. The first few pages should be rewritten so that they start with the broader issues and refine down to the focus of the paper. The reader should know exactly what the paper is about by the second page.

The research question needs to be moved to an earlier location, so as to frame the discussion of the literature. It is currently located at the end of page 5, which is far too late to introduce the focus and scope of the paper. Also, the question you provided on page 5 is very different to the question that you have indicated in the abstract.

Literature review:

You have not identified this as a separate section, however, it appears to have commenced reviewing the literature at line 128. More headings are required to differentiate the different sections, so you should put the heading “literature review” in about here. This would create a natural break point which could be preceded by a statement of the research question and study focus.

You provided a good coverage of the empirical literature, but very little discussion of theories of marketing circular economy organic products (there is currently one paragraph related to this). You must consider the current state of the theory in relation to your following analysis to justify why your study makes a required contribution to the literature.

Method:

This needs to be more thoroughly explained. In particular, whilst you claim to have drawn your measurement items from the literature, evidence supporting their use, such as reference to specific indexes which is connected to the research question is required. More details of the data collection process, including the sample frame, were needed to explain why this was the correct way to answer the research question. At present the sample is not strongly related to the problem.

Analysis:

Cluster analysis was conducted reasonably, although it was unusual to use a statistical analysis to indicate the means, rather than reporting the coefficient and applying the standard tests, such as the cophonetic correlation coefficient which demonstrates that the difference between the coefficients indicates different clusters. Good description of the demographic characteristics of the four groups. If your earlier theoretical framing had developed some key themes that you are investigating (i.e. you had inserted theory into the analysis), you would have been able to compare the nature of the clusters to these important indicator variables. You have more or less done this for the variable of trust, however, the definition of trust varies from cluster to cluster and your assessment of the function of trust in the different clusters seems almost accidental rather than the result of the critical analysis.

I do not think that you can argue that buying locally or from small farmers is a valid indication of a preference for a circular economy. Maybe this represents propensity towards accepting circular economy values, however, the measures are not evidential for this relationship and it does not seem to be strongly connected to the purpose of the paper.

Author Response

Dear Reviewer3,
Thank you very much for your helpful comments. Based on your suggestions, we have reorganized the content to a considerable extent and the individual chapters have been rewritten. We changed the logical order and put the content units together much more strongly. Detailed answers to the comments can be found here:

Comment1:

"Your introduction moves too much around between marketing circular economy to a particular group of customers, operations improvement, economic impact and sustainability goals. All are relevant, however, the sequence of argument is jumbled and discursive, making the theoretical position of the paper unclear. The first few pages should be rewritten so that they start with the broader issues and refine down to the focus of the paper. The reader should know exactly what the paper is about by the second page.

The research question needs to be moved to an earlier location, so as to frame the discussion of the literature. It is currently located at the end of page 5, which is far too late to introduce the focus and scope of the paper. Also, the question you provided on page 5 is very different to the question that you have indicated in the abstract."

Corrections1: 

The introduction chapter has been significantly shortened. In the introduction section, we put the research question and the purpose of the research was precisely identified.
"The significance of focusing on activities that represent local life is reinforced by the view that different strategies towards reaching 'circularity' are already in progress in the home. In the circular economy, consumer collaboration is an essential part; more active involvement of consumers while at this time is still not enough due to lack of environmental awareness and interest in CE. It is an interesting question that organic food consumers can be involved in the circular economy stemming from circular systems? "
Based on your suggestion, we have prepared a separate literature review so that the logical sequence can be well followed within the chapter and between the chapters.

Comment2:

"Literature: You have not identified this as a separate section, however, it appears to have commenced reviewing the literature at line 128. More headings are required to differentiate the different sections, so you should put the heading “literature review” in about here. This would create a natural break point which could be preceded by a statement of the research question and study focus.
You provided a good coverage of the empirical literature, but very little discussion of theories of marketing circular economy organic products (there is currently one paragraph related to this). You must consider the current state of the theory in relation to your following analysis to justify why your study makes a required contribution to the literature.

Corrections2:

Based on your suggestion we have added a new chapter on independent literature processing. New texts are marked with red letters throughout the document. The first part of the thesis was completely rewritten. Some of the circular principles are presented in the literature review, but the purpose of the paper is not to present the building blocks of circular systems. A further 26 new publications were added to the references and several references were omitted.

Comment3:

"Method: This needs to be more thoroughly explained. In particular, whilst you claim to have drawn your measurement items from the literature, evidence supporting their use, such as reference to specific indexes which is connected to the research question is required. More details of the data collection process, including the sample frame, were needed to explain why this was the correct way to answer the research question. At present the sample is not strongly related to the problem."

Corrections3: 

We used the aspects discussed in the literature review especially the aspects used by Kiss et all, 2020 to analyse the receptivity of the circular values among the Hungarian consumers. The data sampling was thoroughly described and it was a general survey focusing on the aspects of trust between consumers and producers, the health-consciousness of the consumers and the use of the direct channels which are contributed to the loops and forming of loops in CE. In the Hungarian food economy, there aren’t any example to investigate so we can analyse only the receptivity of CE values from the side of the consumers.

Comment4: 

"Analysis: Cluster analysis was conducted reasonably, although it was unusual to use a statistical analysis to indicate the means, rather than reporting the coefficient and applying the standard tests, such as the cophonetic correlation coefficient which demonstrates that the difference between the coefficients indicates different clusters. Good description of the demographic characteristics of the four groups. If your earlier theoretical framing had developed some key themes that you are investigating (i.e. you had inserted theory into the analysis), you would have been able to compare the nature of the clusters to these important indicator variables. You have more or less done this for the variable of trust, however, the definition of trust varies from cluster to cluster and your assessment of the function of trust in the different clusters seems almost accidental rather than the result of the critical analysis."

Corrections4:

During the examinations we have software application limitations. We had access only on SPSS basic program, we couldn’t run the cophonetic correlation. The aim of rotation in factor analysis is to better clustering variables in few factors, it showed the strongest correlations in the first vectors and this was an advantage for clustering. We found it helpful in order to give a hint on how to aggregate the date logically before the clustering. We estimated factor score for each respondent, cronbach alpha was high enough and then we loaded cluster analysis. Similar Principal Component Analysis with Rotation Method: Varimax with Kaiser Normalization are done in the paper of Lakatos at al., (2018) but they were placed in the appendix part.
Lakatos, E.S.; Cioca, L.-I.; Dan, V.; Ciomos, A.O.; Crisan, O.A.; Barsan, G. Studies and Investigation about the Attitude towards Sustainable Production, Consumption and Waste Generation in Line with Circular Economy in Romania. Sustainability 2018, 10, 865.

Taken from the paper:
"(In circular economy, consumer collaboration is an essential part; this calls for a new and more active involvement of consumers, while at the same time is still not enough due to lack of awareness and interest in CE. Whether organic food consumers can be engaged in the circular value chain, or who would be the future consumers of food products stemming from circular systems is an interesting topic of research. The important question is, who will choose healthy, more reliable but expensive food in the future? Are sustainable consumption habits independent of age or IT knowledge?)
The group was differentiated by trust (trust in food sold in food stores, safer food can be bought on market), consciousness (label, ingredients, organic food), origin (preference toward Hungarian food, small farmers). 

The group was differentiated by trust (trust in food sold in food stores, safer food can be bought on market), consciousness (label, ingredients, organic food), origin (preference toward Hungarian food, small farmers)."

Comment5:

“I do not think that you can argue that buying locally or from small farmers is a valid indication of a preference for a circular economy. Maybe this represents propensity towards accepting circular economy values, however, the measures are not evidential for this relationship and it does not seem to be strongly connected to the purpose of the paper.”

Corrections5:

Extended explanation is here:

“The impact of local food to the environment is questionable if we think of that the consumers travelling to the site of the farmers. This problem was mentioned also by Kiss et al. (2019). Maybe in this case the requirement of 0-miles is hurt but it fulfils the circular requirements because the consumer would like to know the product the production circumstances consciously. In case of organic food sometimes it can come up against difficulties. In the case of the global organic food market, consumption is separated, and located far from production. However, consumption of organic food has potential in the circular economy (CE) because short food supply chains are connecting to organic food production which are used by the consumers efficiently and represent locally locked production. With the help of innovative solutions of CE applied in organic farming and other environmental - friendly production systems the old bad conditioning (fear from the fake organics) can be avoided.“ Small farmer preference has social responsibility relation and also product quality relation."

In closing, thank you very much for your valuable comments. Based on your proposals, the content of the thesis has been importantly expanded. The quality of the article changed significantly after the corrrections.
The Authors